# An Automatic Defect Detection System for Petrochemical Pipeline Based on Cycle-GAN and YOLO v5

**DOI:** 10.3390/s22207907

**Published:** 2022-10-17

**Authors:** Kun Chen, Hongtao Li, Chunshu Li, Xinyue Zhao, Shujie Wu, Yuxiao Duan, Jinshen Wang

**Affiliations:** 1Petroleum Engineering School, Southwest Petroleum University, Chengdu 610050, China; 2Tianjin Petrochemical Equipment and Instrumentation Research, Tianjin 300270, China; 3Tianjin Petrochemical Corporation, Tianjin 300270, China; 4School of Astronautics, Beihang University, Beijing 100191, China

**Keywords:** petrochemical pipeline, defect detection, Cycle-GAN, YOLO v5, attention mechanism

## Abstract

Defect detection of petrochemical pipelines is an important task for industrial production safety. At present, pipeline defect detection mainly relies on closed circuit television method (CCTV) to take video of the pipeline inner wall and then detect the defective area manually, so the detection is very time-consuming and has a high rate of false and missed detections. To solve the above issues, we proposed an automatic defect detection system for petrochemical pipeline based on Cycle-GAN and improved YOLO v5. Firstly, in order to create the pipeline defect dataset, the original pipeline videos need pre-processing, which includes frame extraction, unfolding, illumination balancing, and image stitching to create coherent and tiled pipeline inner wall images. Secondly, aiming at the problems of small amount of samples and the imbalance of defect and non-defect classes, a sample enhancement strategy based on Cycle-GAN is proposed to generate defect images and expand the data set. Finally, in order to detect defective areas on the pipeline and improve the detection accuracy, a robust defect detection model based on improved YOLO v5 and Transformer attention mechanism is proposed, with the average precision and recall as 93.10% and 90.96%, and the F1-score as 0.920 on the test set. The proposed system can provide reference for operators in pipeline health inspection, improving the efficiency and accuracy of detection.

## 1. Introduction

As the most commonly used component in modern industrial production, the industrial pipeline is one of the most frequently damaged sections in many industrial accidents, including corrosion, scaling, cracking, perforation, and other issues that seriously threaten the service safety of pipelines due to their features of numerous points, extensive areas, enormous volumes, changing curvature, and difficult working conditions. Safety events that resulted from failing to respond to anomalous pipeline conditions in a timely manner have been reported all over the world, so periodic maintenance and inspection are crucial for keeping the pipeline operating steadily for a long time. Fixed-point monitoring and manual testing from external pipelines are the primary methods at present. However, it is required to erect scaffolding, remove external thermal insulation, and scrape rust from the exterior walls when testing, which makes the test results not accurately reflect the whole situation, provides a safety risk during construction, and brings higher construction costs. In addition, the diversity of process medium, complexity of pipeline structure, and low accessibility such as a suspended or buried pipeline will lead to missed detection and false detection.

With the rapid development of pipeline intelligent detection robot technology, pipeline crawling robots have been applied widely to improve pipeline detection efficiency recently. A novel method named Closed Circuit Television (CCTV) [1] which uses a small crawling robot to shoot inside the metal pipe and identify the defects reflected in the video has become the most widely used detection method in engineering fields. As an important supplement to the external detection technology, the in-pipeline detection is carried out based on CCTV to collect the information of the condition in pipelines, and the detection data are transmitted to the intelligent detection platform for video inspection and defect detection. It is challenging to obtain ideal detection results only relying on traditional image detection methods due to the complicated types of pipeline defects and the varied background of video data. Therefore, the current pipeline defect detection work still needs to rely on the professionals to assess and examine the situation in the pipeline. The main shortcomings of the manual work are as follows:(1)Time-consuming and inefficient: It takes interpreters a long time to work on a large number of detection videos;(2)High false and missed detection rate: Even professionals can cause about 20% of false and missed detection due to long-term fatigue, misjudgment, operational errors, and missing data;(3)Non-uniform evaluation results: The interpretation results are heavily influenced by subjective factors because the definitions of some defects are relatively close, or even a single defect can be interpreted as two or three different types;(4)Inaccurate judgment of defect grade: It is difficult for the interpreter to measure the grade of the defect from the currently commonly used video information on subjective judgment.

Therefore, it is necessary to develop a method that can automatically identify the defects inside the pipeline, which reduces labor and time costs while improving the efficiency and quality of the detection.

With the advancement of computer technology, artificial intelligent methods such as machine learning and deep learning are playing a role in more fields, and object detection is one of the important application directions. Therefore, the machine learning approaches are naturally applied to defect detection for such as metals, semiconductors, and fruits [2]. Compared with the traditional detection method, the method based on computer vision shows strong advantages and leads to the automation of the detection process, which improves the efficiency and accuracy of detection while saving labor costs. However, there is not much research in the area of intelligent pipeline defect detection due to the complex pipeline image background, a wide variety of defect categories, and the possibility of different kinds of defects existing in the same area.

In this paper, a comprehensive defect detection system with high accuracy and robustness for the pipeline inner wall is proposed. The main work is as follows:(1)To address the issues of distortion, noise, and uneven illumination caused by the hemispherical camera, the distorted pipe wall image is unfolded by geometric coordinate transformation, and the uneven illumination is equalized by a Retinex-based image enhancement method.(2)For the high repetition of adjacent frames of video with contextual coherence, the image stitching algorithm based on SIFT features is carried out on the unfolded pipe inner wall images to obtain a complete and coherent image;(3)To solve the problems of the small amount of sample data and the unbalanced number of defect categories, a sample enhancement strategy based on Cycle-GAN is proposed in which the defective areas are randomly generated at the non-defect pipe walls to obtain pseudo-samples of defect pipe and enrich the sample diversity based on existing data sets;(4)YOLO (You Only Look Once) v5-based object detection network is used to detect the specific defective areas on the stitched pipeline inner wall images and classify the categories. The proposed network is first pre-trained on similar data sets; therefore, the amount of required data is reduced by transfer learning methods. The Transformer attention mechanism is integrated into YOLO v5 to help the network detect the region of interest in images and improve the detection and recognition accuracy.

## 2. Related Works

### 2.1. Pipeline Internal Defect Detection Based on Computer Vision

With the rapid development of robotics, pipeline CCTV detection technology first appeared in the 1950s and was placed on various active pipeline robots for internal pipeline operations. By the middle of the 1990s, CCTV detection technology had been established as the most sophisticated and reliable method of identifying pipeline conditions worldwide [3,4,5,6]. Robots have greatly expanded the detectable range of pipeline interiors and enabled workers to more clearly inspect the inside conditions while determining the health of the pipeline.

Moselhi et al. [7] introduced image processing into the automatic pipeline defect detection and proposed an automated detection system including four modules of image acquisition, image processing, feature extraction and defect classification, which provides a basic framework for a computer vision-based pipeline internal defect detection system. Yang and Su [8] combined morphological opening operation and OTSU thresholding to segment CCTV video frames, and defined four ideal models of typical pipeline defects of pipe wall collapse, pipe body breakage, object insertion, and joint detachment. However, this method has a great limitation that it has strict requirements on the angle of video shooting and environmental lighting. Myrans et al. [9] extracted GIST features from the images and used the random forest algorithm to discriminate defects in the pipeline images. In addition, they used Hidden Markov Model (HMM) and Order Oblivious Filter (OOF) to process additional information between adjacent frames to further improve the prediction accuracy and reduce the false negative rate (FNR), but the method is tedious and not suitable for fast detection. Guo et al. [10] proposed a change detection based method for pipeline video frames with image alignment, histogram matching, image filtering, and then differencing the current frame of the pipeline video from the reference image to obtain the region of interest and performing multilevel determination of the obtained region of interest, which achieves high accuracy but also has a high false detection rate. Yang et al. [8] extracted entropy features, clustering trends, and other features in the image through co-occurrence matrix and wavelet transform to train a classifier based on SVM for effective pipeline defect detection. In addition, many other CCTV-based methods were proposed and some methods for specific pipeline defects are also proposed. Similarly, Ye et al. [11] accomplish the classification of pipe defects by fusing multiple features of pipe images combined with machine learning algorithms. Khalifa et al. [12] identify crack defects present in pipes by morphological methods after noise removal from the original images. This method can check for small cracks that cannot be judged by the naked eye, but the method cannot make judgments about other defects. Halfawy et al. [13] use morphological techniques, threshold segmentation, and median filtering techniques to delineate the region containing pipe defects in the pipe image, and then extract specific features in that region and perform defect detection by a support vector machine classifier. Like the previous method, this method also targets specific defects, such as pipe defects caused by tree root intrusion.

In conclusion, the above process based on computer vision detection method includes three steps: image preprocessing, feature extraction, and classification. These methods are fast and do not require too much of the original data set. However, many methods can only identify specific defects because the feature extraction is designed for a particular defect, which limits the applicability. In addition, there is also the problem of low accuracy and high false detection rate.

### 2.2. Pipeline Internal Defect Detection Based on Deep Learning

With the development of deep learning technology, it has been widely used in video processing [14,15,16,17], water level calculation [18,19], pipeline semantic segmentation [20,21,22], and pipeline defect classification [23,24] in the field of pipeline detection. Hassan et al. [25] adopted AlexNet [26] into pipeline defect detection and used the images edited from pipeline CCTV videos to form the training set for the model. Wang and Cheng combined deep convolutional neural networks (CNN) with fully connected conditional random fields (Dense CRF) to form an end-to-end neural network named DilaSeg-CRF and jointly train the parameters of segmentation layers and CRF layers. Later, many other CNN-based methods have also been proposed. Hassan et al. [25] proposed a convolutional neural network-based defect classification system for CCTV pipeline videos. The pipeline image dataset is manually labeled according to six predefined defect types, and this CCTV pipeline video defect classification system is able to identify pipeline defects in addition to captions on CCTV pipeline videos, which helps to obtain the location information of defects. Meijer et al. proposed a method of cross-validation to eliminate data leakage and reduce the risk of overestimating the classifier performance. The metrics of sensitivity specificity and recall precision are introduced to measure the performance of pipeline defect classifiers. Wang et al. [27] applied Faster R-CNN to pipeline defect detection; this method can assess the severity of defects as well as the condition of the pipe, but cannot operate in real time. The first work of Faster R-CNN network for sewer defect tracking was used by Kumar et al. [17]. This method can obtain the number of defects. However, it requires training two models rather than an end-to-end framework. Kumar et al. [28] further compared the Faster-RCNN model with the YOLO v3 and the Single-Shot Multibox Detector (SSD), but it is not applicable to structural defect detection. Yin et al. [29] used the YOLO v3 network, which covers defect detection, video interpretation, and text recognition functions. Rao et al. [30] used a network combining CNN and non-overlapping windows, which outperformed existing models in terms of detection accuracy; their study also demonstrated that a deeper CNN model can have better performance, but also requires longer inference time. Li et al. [31] used the Strengthened region proposal network (SRPN) network structure, which can evaluate the defect level based on the effective localization of defects, but it cannot be applied to online processing. Klusek et al. [32] used the YOLO v2 network, which covers the functions of defect classification, detection, and segmentation, but the detection results are weak. Dang et al. [33] proposed a Transformer-based defect detection (Defect TR); this method does not require priori knowledge and can obtain better results with a limited number of parameters. However, the robustness of the method is not strong, and it cannot be practically applied. Ma et al. [34] proposed a multi-defect detection system for sewage pipes based on StyleGAN-SDM and fusion CNN, which alleviated the problems of difficulty in data collection and small amount of data. However, only the complete pipeline images were classified, and there is a lack of pixel-level classification results and positioning information for the accurate defective area.

In short, the detection accuracy and efficiency of methods based on deep learning are far higher than that of the traditional methods. However, the training of neural network model generally requires more labor resources to label the CCTV pipeline video anomalies and establish data sets for training and testing. From the essential standpoint, there are huge gaps in the data amount among different categories of pipeline images, which leads to the extreme imbalance and the overfitting of the networks. From the perspective of a single image, the pipeline image may have a lot of noise including the image itself and labels, which makes it difficult to train a perfect model. In addition, the detailed area in the image may have low brightness, and it is difficult to distinguish some defects by naked eyes, which further increases the difficulty of pipeline defection [35,36,37,38].

## 3. Proposed Methods

The data set used in this work is the video taken inside a pipeline of a petrochemical device to be tested using a hemispheric camera. In order to solve the image distortion problem caused by the hemispherical camera shooting in the pipeline, the image is firstly unfolded circularly, and the multi-scale Retinex [39] with chromaticity preservation (MSRCP) [40] is applied to work out the problem of uneven illumination in the image. In order to make full use of the context information in the video, an image stitching approach based on SIFT features [41] is used to stitch adjacent frames to obtain a complete unfolded image of the pipeline inner wall. However, the number of samples in the original data set after stitching is not enough to support the training of the deep learning network because of the limited available data. Therefore, a sample enhancement strategy based on Cycle-GAN [42] is proposed to randomly generate the defect regions on the original data and expand the sample size by 5.5 times. Based on the enhanced dataset, a YOLO v5 target detection network [43] is adopted to detect and identify the defective area, and the attention mechanism is introduced to improve the recognition results. In addition, transfer learning strategy is used to solve the overfitting problem caused by insufficient sample size. The overall flowchart of this paper is shown in Figure 1.

### 3.1. Pipeline Data Preprocessing

#### 3.1.1. Image Unfolding

In order to separate the defective area accurately and restore the distorted image of the inner wall of the pipeline, the original ring image of the inner wall of the pipeline is firstly unfolded.

Because the pipeline data are mainly illuminated by its own light source when shooting, the image of the inner wall of the pipeline shows the features of black corners, darker center, and brighter inner wall. The inner wall, center, and four corners of the pipeline are divided apart by threshold segmentation method, and the noise is partly removed by the morphological method. The circular region is detected by Hough transform to obtain the center coordinate $O_{1}\left(u_{0},v_{0}\right)$ of the pipeline. The outer circumference of the ring is tangent to the edge of the image considering the features of shooting data, so the outer radius of the ring is r=L/2, where *L* is the size of each frame of the image.

For the original ring image *U*, the pixel coordinate system uOv is established with the center point *O* of the image as the origin, as shown in Figure 2a. The pipeline inner wall image is unfolded counterclockwise with the actual center coordinate of the pipeline $O_{1}\left(u_{0},v_{0}\right)$ considered as the center and horizontal right as 0°. The corresponding unfolded rectangular image I is shown in Figure 2b, where the coordinate system xOy is established with the lower left corner *A* of the rectangle as the origin. Let any point *C* on the inner wall of the pipeline be set as C(x,y) in the unfolded image, and $C^{\prime }\left(u,v\right)$ in the corresponding pixel coordinate of the original image, so the function mapping relation needs to be established as $f:C^{\prime }\left(u,v\right)\in U\to C\left(x,y\right)\in I$.

The unfolded result is shown in Figure 2, and the specific unfolding equations are shown in Appendix A.

#### 3.1.2. Luminance Correction

When using the pipe-climbing robot to shoot inside the pipeline of the petrochemical device, the auxiliary lighting device of the robot is applied since there is no lighting facility inside the pipeline and the light is relatively dim, which will lead to uneven illumination and even serious reflection in some areas of the pipe inner wall image taken. In order to reduce the influence of uneven illumination on the image stitching and defect detection results of the inner wall of the pipeline, it is necessary to perform image enhancement. Because of the above issues, the MSRCP algorithm derived from the Retinex algorithm is mainly used to process images with uneven illumination in this paper.

Retinex is a commonly used image enhancement method with the basic theory that the color of an object is determined by its ability to reflect long-wave (red), medium-wave (green), and short-wave (blue) light, rather than the absolute value of the reflected light intensity. The color of objects is not affected by lighting non-uniformity, leading to its consistency.

The original image *S* can be represented as the product of the illumination image *L* determined by external factors and the reflectivity image *R* determined by the object itself, as shown in Figure 3. The algorithm can calculate the illumination image L from the original image and extract the reflectance image *R* to remove the inconsistent illumination achieving the image enhancement. Use the center surround function F(x,y) to extract the illumination component of an unevenly illuminated image as
(1)$$F\left(x,y\right)=\lambda e^{-\frac{x^{2}+y^{2}}{c^{2}}}$$
where *c* is the Gaussian surround scale, and lamda is a scale. The two values satisfy
(2)∫∫F(x,y)dxdy=1,
and the output image is denoted as
(3)$$r\left(x,y\right)=\log R\left(x,y\right)=\log\frac{S(x,y)}{L(x,y)}=\log S\left(x,y\right)-\log\left[F\left(x,y\right)⊗S\left(x,y\right)\right].$$

Extracting the illumination components of the image through multi-scale can not only compress the dynamic range of the image, but also maintain the consistency of the color perception. The output image based on multi-scale is
(4)$$r\left(x,y\right)=\log R\left(x,y\right)=\sum_{i}Weight_{i}\times \left[\log S_{i}\left(x,y\right)-\log L_{i}\left(x,y\right)\right]$$
where $Weight_{i}$ represents the weight corresponding to each scale, with the sum of each scale weight as 1. In the process of enhancement, the color of the local details in the image may be distorted due to the increase of noise, which cannot display the real color of the objects and cause the deterioration of overall visual perception. To solve this problem, a color restoration function *C* is added to adjust the color distortion caused by contrast enhancement in local areas of the image in MSRCR: (5)$$R_{MSRCR_{i}}\left(x,y\right)=C_{i}\left(x,y\right)R_{MSR_{i}}\left(x,y\right)$$
(6)$$C_{i}\left(x,y\right)=f\left[I_{i}^{\prime }\left(x,y\right)\right]=f\left[\frac{I_{i}\left(x,y\right)}{\sum_{j=1}^{N}I_{j}\left(x,y\right)}\right]$$
(7)$$C_{i}\left(x,y\right)=\beta \log\left[\alpha I_{i}^{\prime }\left(x,y\right)\right]=\beta \left\{\log\left[\alpha I_{i}^{\prime }\left(x,y\right)\right]-\log\left[\sum_{j=1}^{N}I_{j}\left(x,y\right)\right]\right\}$$
where $I_{i}\left(x,y\right)$ represents the image of the *i*th channel, $C_{i}$ represents the color restoration function of the *i*th channel to adjust the color ratio of the three channels, f(·) represents the mapping function of the color space, β is the gain constant, and α is the controlled nonlinear intensity.

The hue of the original image is reasonable, but the color bias problem occurs after processing the classic MSRCR algorithm. Therefore, the MSRCP algorithm is used to perform Retinex on the intensity data of the image, and then the data are mapped to each channel according to the original RGB ratio, which enhances the image while preserving the original color distribution.

#### 3.1.3. Image Stitching

The data type used in this paper is video data, which has the features of coherent context information and high similarity between adjacent frames. In order to fully utilize the temporal information in the video data while preventing repeated detection of similar frames, the images of the pipeline inner wall are unfolded and stitched to obtain a coherent and complete image and the defect regions.

Suppose $I_{t}$ represents the video frame unfolding graph at time *t*, and $I_{\left(t+1\right)}$ represents the video frame unfolding graph at time t+1. The feature points should be detected and matched first when stitching two frames of images. In this paper, the scale-invariant feature transform (SIFT) matching algorithm is used to register the unfolded images of two adjacent frames.

SIFT algorithm implements feature matching mainly in the following three processes:(1)Extract key points: Search image locations on all scale spaces, and identify the potential interest points with scale and rotation invariance through a Gaussian differential function.(2)Locate key points and determine feature orientation: A fitted model is used to determine the position and scale at each candidate position that feature sets $\chi_{t}=\left\{\alpha_{1},\ldots ,\alpha_{h}\right\}$ and $\chi_{t+1}=\left\{\beta_{1},\ldots ,\beta_{s}\right\}$ are established from the $I_{t}$ and $I_{\left(t+1\right)}$, respectively.(3)Match feature point pairs to build the correspondence between images: The random sampling consensus algorithm (RANSAC) is adopted to eliminate mismatched key points in this paper. The data are cut into two parts of the correct points as inner points, and the anomalous data as outer points. The iterative method is used to find the optimal parameter model in the data set, in which the points that do not conform to the optimal model are defined as “outer points”. Finally, the subsets ${\hat{\chi }}_{t}=\left\{\alpha_{h_{1}},\ldots ,\alpha_{h_{m}}\right\}\subseteq \chi_{t}$ and ${\hat{\chi }}_{t+1}=\left\{\beta_{s_{1}},\ldots ,\beta_{s_{m}}\right\}\subseteq \chi_{t+1}$ are generated, and for each $k\in \left\{h_{1},\ldots ,h_{m}\right\}$, there exists one $j\in \left\{s_{1},\ldots ,s_{m}\right\}$ such that $\alpha_{k}$ matches $\beta_{j}$.

The homography matrix *H* between the two images is calculated after obtaining the SIFT feature matching point pairs,
(8)$$H=\left[\begin{array}{ccc} R_{a} & R_{b} & T_{x}\\ R_{c} & R_{d} & T_{y}\\ W_{a} & W_{b} & 1 \end{array}\right]$$
where $H_{R}=\left[\begin{array}{cc} R_{a} & R_{b}\\ R_{c} & R_{d} \end{array}\right]$ is the rotation matrix, $M_{T}={\left[\begin{array}{cc} T_{x} & T_{y} \end{array}\right]}^{T}$ is the translation matrix, and $M_{W}=\left[\begin{array}{cc} W_{a} & W_{b} \end{array}\right]$ is the deformation matrix.

Since the used data are from the pipe-climbing robot moving steadily in a straight line in the pipeline, without rolling along the pipeline or view transformation, the calculation can be simplified as the estimation of the translation component $M_{T}={\left[\begin{array}{cc} T_{x} & T_{y} \end{array}\right]}^{T}$ between two adjacent frames of images, and the affine transformation is not applied to the new images in order to prevent extra distortion. Two adjacent frames of images are stitched together after the deformation components are obtained. The complete pipeline inner wall image and defect situation are obtained by continuously stitching new video frames $I_{t+n}$.

### 3.2. Sample Balance Strategy

#### 3.2.1. Framework Structure of Cycle-GAN

In this paper, Cycle-GAN is used to randomly generate pseudo samples in the defective area based on the original data to expand the training data set because the original data set has few samples, and the number of different defect samples is unbalanced between classes.

The main benefit of Cycle-GAN is that it can train the network to transfer picture styles without needing pairs of data [44]. Another cycle-consistent constraint is implemented in the Cycle-GAN network to get over the issue that the network ignores the input and randomly generates the output image due to the lack of paired training data. The input original domain image is transformed in two steps: (1) Mapped to the target domain with the mapping relationship as *G*; (2) returned to the original domain to obtain the secondary generated image with the mapping relationship as *F*. The influence of mismatched training images is eliminated and the quality of the generated images is improved, by ensuring that the secondary generated images match the source images as closely as feasible. The principle of Cycle-GAN is shown in Figure 4.

Cycle-GAN is a ring-shaped mechanism containing two groups of generators *G* and *F* and two groups of discriminators $D_{x}$ and $D_{y}$. The generator *G* converts the image in the original domain into the image in the target domain of image *Y*. In turn, *F* maps the image under the domain of image *Y* back to the domain of image *X*. $D_{x}$ evaluates the reliability of images in the domain of image *X*, while $D_{y}$ is for images in the domain of image *Y*. In this paper, the images of the pipeline inner wall without defect represent the original domain images, while the images of the pipeline inner wall with defect represent the target domain images. Multiple network models are trained for different defect categories.

The generator and discriminator are trained alternately by the Adam optimizer in Cycle-GAN. The loss function is adopted to maximize logD(x,y) during the training of the generator *G*, while the loss function of the discriminator *D* is divided into the average of the loss of original images and generated images.

The loss function of Cycle-GAN consists of three parts with the first two as the loss of general CGAN,
(9)$$L_{GAN}\left(G,D_{Y},X,Y\right)=E_{y}\left[\log D_{Y}\left(y\right)\right]+E_{x}\left[\log\left(1D_{Y}\left(G\left(x\right)\right)\right]\right.$$
(10)$$L_{GAN}\left(F,D_{X},Y,X\right)=E_{X}\left[\log D_{X}\left(x\right)\right]+E_{Y}\left[\log\left(1D_{X}\left(F\left(y\right)\right)\right]\right.$$

Another part of the loss function comes from the definition of cycle consistency loss, as the two mappings of *F* and *G* are required to satisfy F(G(y))≈y and G(F(x))≈x. In addition, the two processes are expected to be mutually inverse that an image can be returned to the original image under two iterations: (11)$$L_{cyc}\left(F,G,X,Y\right)=E_{x}\left[{∥F\left(G\left(x\right)\right)-x∥}_{1}\right]+E_{y}\left[{∥G\left(F\left(y\right)\right)-y∥}_{1}\right]$$

The final loss function is
(12)$$L=L_{GAN}\left(F,D_{Y},Y,X\right)+L_{GAN}\left(G,D_{X},X,Y\right)+\lambda L_{cyc}\left(F,G,X,Y\right)$$
(13)$$G^{*},F^{*}=\arg\min_{G,F}\max_{D_{X},D_{Y}}L\left(G,F,D_{X},D_{Y}\right)$$

#### 3.2.2. Improved Loss Function of Cycle-GAN Based on Gradient Penalty

There are some following problems and challenges in training Cycle-GAN due to its evolution from the ordinary GAN network [45]:(1)It is necessary to carefully design and balance the training degree of the generator and the discriminator in the process of training; otherwise, it serious gradient disappearance in the generator will occur when the discriminator is trained better, making it difficult to continue training;(2)The loss function of the generator and the discriminator cannot indicate the training process, due to the lack of meaningful metrics associated with the quality of generated images;(3)The generated images may lack diversity because of the mode collapse. These defects are caused by two reasons: one is that the distance measurement type of the equivalent change is unreasonable, and the other is that the samples generated after random initialization by the generator have almost no non-negligible overlap with the real sample distribution.

The main idea of the improvement of Cycle-GAN is to introduce the gradient penalty term, with the use of Wasserstein distance in place of JS divergence as a part of the loss function. The normalization method in Cycle-GAN is instance normalization from the idea of the least squares GAN network (LSGAN) [46], so only two steps are made when improving: (1) The loss functions of the generator and the discriminator are no longer logarithmic, and the gradient penalty term is added to the discriminator loss function; (2) The sigmoid layer is removed from the last layer in the discriminator because it needs to approximate the Wasserstein distance and performs a regression task instead of a binary classification task.

The improved loss function is
(14)$$L_{WGAN-GP}\left(F,D_{X},Y,X\right)=E_{y}\left[D_{X}\left(y\right)\right]-E_{x}\left[D_{X}\left(G\left(x\right)\right]\right.$$
(15)$$L_{WGAN-GP}\left(G,D_{Y},X,Y\right)=E_{x}\left[D_{Y}\left(G\left(x\right)\right]-E_{y}\left[D_{Y}\left(y\right)\right]\right.$$
(16)$$L=L_{WGAN-GP}\left(F,D_{Y},Y,X\right)+L_{WGAN-GP}\left(G,D_{X},X,Y\right)+\lambda L_{cyc}\left(F,G,X,Y\right)$$

### 3.3. Defect Detection Based on YOLO v5

In this paper, an object detection method based on deep learning is performed to detect and identify the defect of the pipeline inner wall after stitching the images of the inner pipeline wall. Object detection is to find and classify one or more objects of interest in images, and determine their category and location, which is often used in autonomous vehicles, face detection, and other fields. In this paper, the object detection algorithm of YOLO v5 is applied to the task of pipeline surface defect detection and obtains gratifying results.

#### 3.3.1. Network Structure of YOLO v5

Object detection has made great breakthroughs in recent years. Most popular approaches can be divided into two categories: one is the R-CNN series algorithms based on region proposal (R-CNN [47], Fast R-CNN [48], Faster R-CNN [49], etc.), which belong to two-stage algorithms in which the candidate box is first generated as the target position, where the classification and regression are performed. The two-stage approach has high recognition accuracy but slow speed. The other is a one-stage algorithm such as YOLO and SSD [50], in which only a backbone convolutional neural network CNN is used to directly predict the categories and locations of different targets. One-stage approach is faster, while most recognition accuracy is lower.

In this paper, YOLO v5 network is adopted to identify the defect in the pipeline inner wall, which is an object detection method proposed by the ultralytics team in May 2020. YOLO v5 has greatly improved the detection accuracy while retaining the advantage of fast detection speed compared with the previous versions of the YOLO network, which mitigates the drawbacks of the poor detection accuracy of the one-stage network.

The core idea of YOLO is to input the entire image into the network which is similar to Faster-RCNN, and directly regress the locations and categories of the bounding boxes at the output layer. The specific process is as follows:(1)The image is divided into S×S grids cells, utilized for predicting the objects when the centers fall in the grids;(2)*B* bounding boxes are predicted per grids cell, with (x,y,w,h) and confidence for a total of five values predicted on each bounding box.(3)The category is predicted in each grid cell at a total of *C* categories. The final output tensor is the S×S×(5×B+C) dimension, which is the prediction of the locations and categories for B bounding boxes of S×S grids cells. The network structure of YOLO v5 is shown in Figure 5.

(1) Input

(a) Mosaic data enhancement

The mosaic data enhancement is operated on the data set in which four input pictures of the pipeline inner wall in random scaling, random cropping, and random arrangement are stitched at the input end of YOLO v5, which increases the size diversity of pipeline images and improves the robustness of detection of small inner wall defective areas.

(b) Adaptive anchor box

YOLO network outputs the prediction boxes on the basis of the initial anchor boxes which are set with different lengths and widths for different data sets. The prediction boxes are compared with the ground truth for calculating the gap between the two and updating the network parameters by back-propagation to fit the shape of the defective area of the pipeline.

(2) Backbone

(a) Focus layer

The input images are sliced in interval pixels to obtain four images halved in size and complementary content in the Focus layer before entering a backbone network, which concentrates the information of width and height space in the channel space. Finally, the new image is convolved again to produce the double down-sampled feature map without information loss.

(b) CSP structure

CSPDarknet53 is used as a backbone structure in YOLO v5, which is a backbone structure of five CSP modules based on the YOLO backbone network of Darknet53 and the ideas of CSPNet. The size of the convolution kernel in front of each CSP module is 3 × 3 with the stride of 2 for down sampling. The CSP module first divides the feature map of the base layer into two parts and merges them through the cross stage approach, which can reduce the computation flops while ensuring the accuracy. Therefore, the CSP network structure in the backbone network of YOLO v5 has the advantages of enhancing network learning ability, reducing computing bottleneck and memory cost.

(3) Neck

FPN+PAN structure is adopted in the Neck of YOLO v5 in which the FPN (Feature Pyramid Network) layer conveys strong semantic features from the top down, while the PAN (Path Aggregation Network) layer conveys strong localization features from the bottom up. Feature aggregation is performed on different detection layers from different backbone layers to improve the ability of feature extraction.

(4) Output terminal

(a) Loss function of bounding box

$CIoU_{L}oss$ is used as the loss function of the bounding box, considering the coverage area, center point distance, and aspect ratio between the prediction box and the bounding box, respectively,
(17)$$L_{CIoU}=1-IoU+\frac{\rho^{2}\left(b,b^{gt}\right)}{c^{2}}+\alpha v^{2}$$
where *v* is the distance between the aspect ratios of B and G,
(18)$$v=\frac{4}{\pi^{2}}{\left(\arctan\left(\frac{w^{gt}}{h^{gt}}\right)-\arctan\left(\frac{w}{h}\right)\right)}^{2}.$$

α is a weight coefficient,
(19)$$\alpha =\frac{v}{(1-IoU)+v}$$

(b) non-maximum suppression

After detecting and classifying the defective areas, the non-maximum suppression (NMS) is used to screen the detection boxes to solve the problem of overlapping target boxes. The IoU value is calculated between the detection boxes with the highest confidence and other detection boxes, and the boxes with the IoU value exceeding the specified threshold are suppressed. The above process is iterated until each object owns only one corresponding detection box. In YOLO v5, DIoU is introduced to quantify the overlapping parts, and $DIoU_{NMS}$ is used to screen the detection boxes, improving the detection of overlapping defective areas.

#### 3.3.2. Transformer Self-Attention Module

Attention mechanisms originate from the study of human vision in which humans selectively focus on a portion of all information while ignoring other visible information due to bottlenecks in information processing. Neural attention mechanisms allow neural networks to focus on subsets of their inputs or features by selecting specific inputs. Attention can be applied to any type of input regardless of the shape, which is a resource allocation scheme to solve the problem of information overload in the case of limited computing power because it allocates computing resources to more important tasks.

The encoder part of Vision Transformer [51] is introduced in the replacement of the bottleneck part in the original C3 module in this paper, mainly including MSA [52] and MLP [53]: (20)$$z_{0}=\left[x_{\mathrm{class}\;};x_{p}^{1}E;x_{p}^{2}E;\ldots ;x_{p}^{N}E\right]+E_{pos},\;E\in R^{\left(P^{2}\cdot C\right)\times D},E_{pos}\in R^{(N+1)\times D}$$
(21)$$z_{l}^{\prime }=MSA\left(LN\left(z_{l-1}\right)\right)+z_{l-1},\;l=1,\ldots ,L$$
(22)$$z_{l}=MLP\left(LN\left(z_{l}^{\prime }\right)+z_{l}^{\prime },\;l=1,\ldots ,L\right.$$
(23)$$y=LN\left(z_{L}^{0}\right)$$

MSA module includes three parts of multi-head self-attention, skip connections (Add), and layer normalization (Norm), while the MLP module includes three parts of feedforward network (FFN), skip connections (Add), and layer normalization (Norm). An MLP is used as the classification head behind layer normalization.

The transformer encoder block increases the ability to capture different local information, and also makes use of the self-attention mechanism to mine the potential of feature representation, which improves the detection of the network on a small-scale defect.

## 4. Implementation Details and Results

### 4.1. Pipeline Data Preprocessing

#### 4.1.1. Analysis of the Pipeline Dataset

The pipe-climbing robot equipped with lighting source and camera equipment is used to move forward axially inside the pipe to be inspected with a traveling speed of 3 m per minute. The robot can record 180 m of video per hour with a frame rate of 15 fps, and the resolution of each frame is 2888 × 2888 pixels. The defective area of the pipeline inner wall is classified and labeled according to the inspection report given by professional pipeline maintenance personnel. There are mainly four types of surface defects in the used pipeline video data, and examples of different categories are shown in Figure 6.

a. Corrosion: a local corrosion form of small holes in the corrosive medium on the metal surface, which can seriously cause the perforation damage of the pipeline;

b. Oxide layer peeling: the oxide layer is prone to falling off when the pipeline is heated at high temperature, which can lead to a pipe burst accident in severe cases;

c. Sediment: pipeline blockage and extraneous matter deposition and other issues found in the inspection process;

d. Penetration: The penetration of objects caused by inherent structure or defect needs to be identified, and the pipe-climbing robots must be stopped moving forward in time.

In this paper, the original data set consists of 21 videos shot during the maintenance of the pipeline, including drainage pipes, distillation unit pipes, catalytic fractionation tower pipes, and reforming collection pipes, with a total duration of about 1170 min. In order to convert the video data into an image data set, one frame is taken every 15 frames as one second in videos, with a total of about 35,000 images selected for subsequent processing operations of image unfolding, luminance correction, and image stitching. In addition, 706 images are obtained as the original data set after stitching of adjacent frames, including 1015 places of corrosion, 603 places of oxide layer off, 448 places of accumulation, and 55 places of insertion.

Figure 7 shows the video frame $U_{t}$ and the unfolded image of the pipeline inner wall $I_{t}$ at time *t*. Considering that the most serious deformation appears on the outermost and closest areas to the center in $U_{t}$, the 0.8r width area in the middle of the ring region is selected for expansion.

#### 4.1.2. Preprocessing Results

Balanced illumination is applied to the unfolded image $I_{t}$ by three image enhancement methods derived from the Retinex method including MSRCR, MSRCP, and auto-MSRCR. The processing results of the three approaches are compared in Figure 8.

Since the image enhancement method for balancing the influence of illumination is mainly for the image stitching, the focus is whether the brightness of the whole pipeline inner wall image is at the same level, making the seams of subsequent image stitching transitioning more natural. As shown in Figure 8, the result images of MSRCR lose more color information, although the illumination is relatively average. The images processed by Auto-MARCR retain good color and texture information; however, the brightness gap between the upper and lower regions of the image is too large, which is not suitable for image stitching. MSRCP has a good performance in processing image details and brightness average. Therefore, MSRCP is adopted to remove the uneven illumination of the image of the pipeline inner wall in this paper.

The enhanced images will be applied to stitching the inner wall of the pipeline, with the stitching results shown in Figure 9 for continuous $I_{t},I_{t+1},\ldots ,I_{t+n}$ frames. It can be seen in Figure 9a that the integral stitched image without image enhancement shows striped noise interference due to the obvious brightness gaps at the seams, while the seam transition of the stitched image after image enhancement is natural without an obvious stitching trace in Figure 9b. It can be concluded that the image enhancement method used in this paper balances the illumination well.

### 4.2. Construction of Pipeline Defect Image Database

#### 4.2.1. Defect Generation Based on Cycle-GAN

Directly using the unfolded pipeline data to train the defect detection network will cause overfitting due to the limited amount; therefore, the Cycle-GAN network is adopted to generate new defective areas on the original data set to expand the data set and increase the diversity of samples. Training on the whole image with multiple defect areas of different types will not be able to obtain defect details and detect defective areas accurately for subsequent models; therefore, the non-defective areas of the original dataset and the defective areas of different categories were cut out as the training set for Cycle-GAN.

Defects are mainly divided into four categories: corrosion, oxide layer peeling, sedimentation, and penetration. Among them, the number of penetration defects is too small to fit the requirements for the number of training data of Cycle-GAN and obtain effective generation results. Therefore, non-defective areas are selected to randomly generate new samples of other three types of defects. The three types of local defect images are shown in Figure 10.

The training data set for Cycle-GAN includes 1126 non-defect images, 245 corrosion images, 735 oxide layer peeling images, and 169 sedimentation images, which are uniformly resized to 256 × 256 pixels. The hyperparameters are set as follows: the batchsize is 1, the learning rate is 0.0002, and the decay epoch is 9.

The non-defect images cropped from the original dataset and three different types of defect images generated by Cycle-GAN are shown in Figure 11. It can be seen that the generated images are similar to the real situation, which can be used as samples for training the subsequent defect detection model.

The corresponding defect region *N* is generated by Cycle-GAN on the non-defect region, which is randomly selected on the images of the original dataset. In order to avoid the overlap of the target area and original defective area $M_{i}$, the $IOU_{i}$ values between the target area *N* and $M_{i}$ are calculated, and the generation would be continued only when $\sum_{i=1}^{n}IoU_{i}=1$. The fusion strategy is adopted to fuse the new defect image D(x,y) with the background I(x,y) better after generating the local defect image.The fusion results are shown in Figure 12, and the fusion equation is as follows: (24)$$\alpha =\frac{\left|x-\frac{w}{2}\right|}{\frac{w}{2}},\beta =\frac{\left|y-\frac{h}{2}\right|}{\frac{h}{2}},\gamma =\sqrt{\frac{\alpha^{2}+\beta^{2}}{2}}$$
(25)N(x,y)=γ·D(x,y)+(1−γ)·I(x,y)
where N(x,y) is the fusion result, α,β,γ are transition parameters, and w×h is the size of the area to be merged.

#### 4.2.2. Pipeline Data Set for Defect Detection

The original data set obtained 706 images, including 1015 samples of corrosion, 603 samples of oxide layer off, 448 samples of accumulation, and 55 samples of insertion. After data enhancement, a total of 3200 images are generated by Cycle-GAN as the generated data set, which includes 3565 samples of corrosion, 2797 samples of oxide layer peeling, 3627 samples of sediment, and 3124 samples of penetration. The distribution ratio of original and generated samples for different types of defect is shown in Figure 13, which indicated that the number of each type is balanced, and the total number of samples is sufficient to meet the subsequent training requirements for the detection model.

The augmented pipeline data set for the defect detection model contains 3906 images in total as about 5.5 times larger than the original data set, including 4580 corrosion areas, 3400 oxide layer peeling areas, 4075 sediment areas, and 3179 penetration areas. The augmented data set is divided into three parts of training, verification, and testing in the ratio of 6:2:2, where the original and generated images are also distributed in a ratio of 6:2:2 in the three parts. The proportion of the sample distribution is shown in Figure 14.

### 4.3. YOLO v5 Training and Evaluation

#### 4.3.1. Computer Configuration

Training deep learning networks require powerful hardware support. All the training and generation were implemented on a GPU server with Nvidia A40 4×48 GB GPU, Linux operating system, CUDA 11.1, Python 3.8, and Pytorch1.8.0.

#### 4.3.2. Transfer Training

Transfer learning methods are frequently used to transfer prior knowledge learned in one field to the other similar fields when deep neural networks lack sufficient data support. In this paper, the amount of videos captured by the pipe-climbing robot is small, and the information on various types of defect is scarce. Therefore, the transfer learning method is introduced to train the feature extraction ability for the network on similar data sets and transfer the knowledge of target detection to the pipeline defect detection task.

The steel surface defect detection competition data set released by Kaggle in 2019 is used as the pre-training data set in this paper. In addition, 9500 images of the steel data set are selected to train the defect detection network, which are divided into four different defect categories, and each image has 1–2 different defect categories.

The specific training strategies is as follows: (1) The whole network weights are trained for 500 epochs on the steel data set to obtain a pre-trained model, with the data set divided into 7000, 1000, and 1500 images for training, verification, and testing, respectively. (2) Based on the pre-training model, the classification prediction head is randomly initialized and firstly trained for 70 epochs on the expanded data set with the network parameters of the backbone frozen to increase training efficiency and remain the feature extraction network. The whole model of all weights are subsequently trained for 30 epochs to fine-tune the network parameters and obtain the final model results.

#### 4.3.3. Hyperparametric Evolution

The hyperparameters in deep neural network are the external variables that need to be manually set and adjusted continuously to seek the optimal combination because the model changes with the change of hyperparameters.

In this paper, the method based on Genetic Algorithm (GA) provided by YOLO v5 to optimize the hyperparameters. GA is a computational model of biological evolution which simulates the natural selection and genetic mechanism of Darwin’s biological evolution, which searches the optimal solution by simulating the natural evolution process. The default parameters are the hyperparameters learned by training on the coco data set, and the hyperparameters are set to evolve once every 50 training epochs for a total of 100 evolutions. As a result, the optimal combination of hyperparameters is obtained in the 93rd pass, as the evolution process shown in Figure 15. The x-axis indicates the value of hyperparameters, the y-axis indicates fitness, and different colors indicate the frequency of the results, where yellow indicates a higher frequency. The final hyperparameter settings used for training are as follows: initial learning rate $lr_{0}=0.01199$, final learning rate $lr_{f}=0.05053$, SGDmomentum=0.90091.

The network is trained for 500 rounds using the hyperparameter combination obtained through evolution, with the validation to evaluate the performance and prevent overfitting in each round of iteration. As the training results shown in Figure 16, the detection accuracy, recall and mAP tend to be stable around 100 rounds, and finally the detection accuracy on the validation set is 0.957, and the recall is 0.907.

#### 4.3.4. Results and Evaluation

Based on the various corresponding relationship of the prediction result and ground truth, the prediction result for the classification task is separated into four categories: TP (true positive), TN (true negative), FP (false positive), and FN (false negative). In this paper, the precision, recall, and F1 score of the detection model for various types of defects are calculated to assess the defect detection performance.

The precision rate represents the ratio of true positive samples to predicted positive samples, which measures the accuracy of the detection results. The recall rate represents the ratio of samples predicted to be positive among all samples with a true positive value, which measures the comprehensiveness of the detection results: (26)$$\mathrm{Precision}=\frac{\mathrm{TP}}{\mathrm{TP}+\mathrm{FP}}$$
(27)$$\;\mathrm{Recall}\;=\frac{\mathrm{TP}}{\mathrm{TP}+\mathrm{FN}}$$

Both the accuracy of classification and the comprehensiveness of detection should be taken into account in the classification problem. Therefore, F1 score is used to evaluate the model by integrating precision and recall: (28)$$F1\text{-}\mathrm{score}=\frac{2}{\frac{1}{\;\mathrm{Precision}\;}+\frac{1}{\;\mathrm{Recall}\;}}=\frac{2\cdot \;\mathrm{Precision}\;\cdot \;\mathrm{Recall}\;}{\;\mathrm{Precision}\;+\;\mathrm{Recall}\;}$$

The detection results are displayed in Table 1. Each column represents the true category, and the sum of each column represents the total number of defects in that category; each row represents the predicted category, and the sum of each row represents the total number of the defect prediction in that category. The average detection time of each frame is 0.074 s, the precision of all types of defect detection is above 90%, the average recall rate is 90.67%, and the F1 score is above 0.9 in this paper, which indicates that our proposed method can basically detect all the defective areas of the pipeline inner wall while ensuring the accuracy, and can provide a strong guarantee for the pipeline health detection.

In the four types of defects, the recall rate of oxide layer peeling is low with some missed detections, which may be the small gap between the peeling oxide layer and the background part that causes some defective areas not detected. In addition, the classification error cases often occur between the corrosion and sediment due to the high similarity of the two types. The network prediction part needs to be further trained to improve the detection accuracy and recall.

To evaluate the detection results of the defect detection model on the original dataset, we use the model to detect defect regions on the original dataset and the generated dataset separately, and the results are shown in Table 2 and Table 3. The average accuracy of detection on the original image is 90.07% and the average recall is 89.67%, which is similar to the result on the integrated dataset, indicating that the defect detection model proposed in this paper has the capability of practical application. However, the performance on the original image is slightly inferior to that on the integrated data set because the background of the original image is more complex and the transition at the edge of defective area is blurred, making it difficult to delineate the exact defective area.

As seen in Table 2 and Table 3, the detection results of the defect detection model on the original and generated images are similar, indicating that the generated images are clear, recognizable, and similar to the original images, and can be applied to train the defect detection network. However, since the generated defect images are more regular in shape, the accuracy and recall of detection are slightly higher than those of the original images, which reduces the robustness training of network. The credibility of the generated images can be further enhanced by improving the fusion strategy in the future.

## 5. Discussion

### 5.1. Comparison with Different Preprocessing Strategies

In order to evaluate the effectiveness of the preprocessing strategy, the detection results of the proposed preprocessing method were compared with another two versions of data set: one is the original ring images without unfolding, and the other is the unfolding images but without stitching. In addition to different preprocessing strategies, the subsequent detection networks are consistent including the YOLO v5-based defect detection model and the transfer learning strategy.

The defect detection results based on original ring images without unfolding are shown in Table 4. The missed detection in the experiment is mainly the small-scale defective area near the depth of the pipeline. The image distortion caused by the shooting angle compressed the size of the defective area deep in the pipe, and furthermore, the image quality degradation caused by the uneven illumination resulted in some missed detections.

The defect detection results based on the unfolded but not stitched images are shown in Table 5. Some defective areas in unstitched images are incoherent, which results in missed and false detection, as well as inaccurate classification. Additionally, the defect detection model trained on the unstitched data set has a considerable gap in the detection results of different categories, due to the imbalance among the various sample types. In particular, the detection accuracy of sediment, which includes a small amount and insufficient features, is a little lower than that of corrosion and oxide layer peeling.

The detection results of our proposed method and the two preprocessing methods mentioned above are shown in Table 6. Compared to the detection results on original data directly, the proposed data processing method improves detection precision by 10.79%, recall by 6.1%, and F1-score by 0.084. While compared to the results on the unstitched images, the proposed data processing method improves detection precision by 7.59%, recall by 3.15%, and F1-score by 0.054. The comparison results clearly demonstrate that the proposed pre-processing method of unfolding and stitching with balanced illumination can significantly improve defect detection in this paper.

### 5.2. Data Enhancement by Cycle-GAN

The results of verifying the effectiveness of the sample augmentation strategy are shown in Table 7. The data set without data enhancement is divided into three parts of training, validation, and testing in the ratio of 8:1:1, which is fed into the head part of YOLO v5 on the basis of the pre-training model with the same training strategy as the proposed method. The detection result on the test set without data enhancement is shown in Table 6, with the average precision rate of 43.20%, the average recall rate of 52.53%, and the F1-score of 0.474. An overfitting phenomenon occurred in about 30 rounds during network training, leading to the poor detection result on the test set, which indicates that the network detection performance cannot meet the standards for the practical application.

Table 2 and Table 7 show the detection results on original data using the augmented dataset and the original dataset as training sets separately. Comparing Table 2 and Table 7, the network model shows a significant improvement in detection performance after adding the generated images in the training set. In the augmented training set, the generated samples account for a larger proportion. The improvement of the model performance indicates that the generated samples have high similarity to the original samples, which improves the network feature extraction ability and robustness in complex environments.

### 5.3. Comparison with Different Attention Mechanisms

To test the effectiveness of the Transformer attention mechanism, three different defect detection models based on YOLO v5 are trained with the same data set and training strategy for comparison: (1) the original YOLO v5 model; (2) the YOLO v5 model with CBAM attention mechanism added to the backbone; and (3) the YOLO v5 model with SE attention mechanism added to the backbone. The detection results of the three methods above are shown in Table 8, Table 9 and Table 10, respectively, and compared with the proposed detection model in Table 11. Compared to the original YOLO v5 model without an attention mechanism, the precision of our proposed detection model is improved by 2.53%, the recall is improved by 2.66%, and the F1-score is improved by 0.026; when compared to the model with CBAM, the precision of our model is improved by 1.75%, the recall is improved by 1.70%, and the F1-score is improved by 0.017, while, compared to the model with SE, the precision is improved by 2.03%, the recall by 1.73%, and the F1-score by 0.018. The difference of detection results is greatest in the small-scale defect detection, which demonstrates that the method incorporated with the Transformer attention mechanism can improve the detection of small-scale defects.

### 5.4. Robust Analysis

Robustness refers to the ability of the model to maintain perfect detection results in different scenarios, as an important metric of whether the model can be put into practical application. Figure 17, Figure 18, Figure 19 and Figure 20 shows the detection results of the proposed method on different types of pipeline images. For the distillation pipeline in Figure 17, corrosion is the most common type of defect and difficult to detect due to the large scale change and complex background, while the proposed method detects defective areas of various scales well with the precise region division. For the catalytic pipeline in Figure 18 with mainly corrosion and sediment as the defects, the proposed method can maintain accurate detection performance even for small-scale sediment. For the reforming collection pipeline in Figure 19, oxide layer peeling is the most common type of defect with a vague boundary and obscure features, while the proposed method accurately divides the defective area in the bounding boxes. In Figure 20, the drainage pipeline is dominated by the sediment of oil pollution with a large coverage area, which makes it difficult for accurate detection. However, as a result, the proposed method can detect all the oil pollution coverage. Therefore, the proposed method can maintain satisfactory detection performance in different pipelines and complex environments, and has strong robustness and high generalization ability.

## 6. Conclusions

The petrochemical pipeline is an important transportation tool in industrial production, while the inner wall of the pipeline has different degrees of defects during service. At present, pipeline health detection mainly relies on manual work causing high cost and low efficiency. In this paper, on the video data set shoot by the pipe-climbing robot, a sample enhancement strategy based on Cycle-GAN and a defect detection system of the pipeline inner wall based on improved YOLO v5 are proposed. The main contributions are summarized as follows:(1)Aiming at the image distortion during data collection, a geometric coordinate transformation method is used to unfold the ring area of the pipe wall. For the uneven illumination caused by robot lighting equipment, the brightness of the image is equalized by the MSRCP algorithm;(2)In order to make full use of the context information in the video data, an image stitching algorithm based on SIFT feature is applied to stitch the continuous unfolded images of the inner wall, obtaining a coherent and tiled image of the pipeline inner wall, which can help the defect detection network to identify the defective area accurately;(3)To address the issues of complex pipeline environment, a small amount of data and unbalanced sample classes, a sample enhancement strategy based on cycle-GAN is proposed in which local defective areas are randomly generated on the original inner wall image. A total of 3200 images are generated as the augmented dataset for defect detection as expanded to about 5.5 times of the original, including 3565 corrosion areas, 2797 oxide layers peeling areas, 3627 sediment areas, and 3124 penetration areas. This data enhancement strategy not only enriches the diversity of samples, but also solves the long-tail problem caused by the imbalance between sample classes.(4)For the pipe wall defect detection task, the YOLO v5-based model is proposed. In order to solve the overfitting problem caused by small samples and large models, the transfer learning strategy is adopted to train the feature extraction network with similar data sets. Aiming at the changing scale of the defective area, the Transformer attention mechanism is integrated into YOLO v5 to help the network find the region of interest in the complex background and improve the detection and recognition of small-scale defects. The average detection precision of the proposed method on the test set is 93.10%, the average recall rate is 90.96%, and the F1-score is 0.920.

The internal environment of the pipeline is complex, and this paper is aimed at detecting the more common defects as corrosion, oxide layer peeling, sediment, and penetration. In the future, we will collect more images of less common defect types and incorporate them into the defect data set of the pipeline inner wall to improve the defect detection system, so as to better serve pipeline health detection.

## Figures and Tables

**Figure 1 sensors-22-07907-f001:**
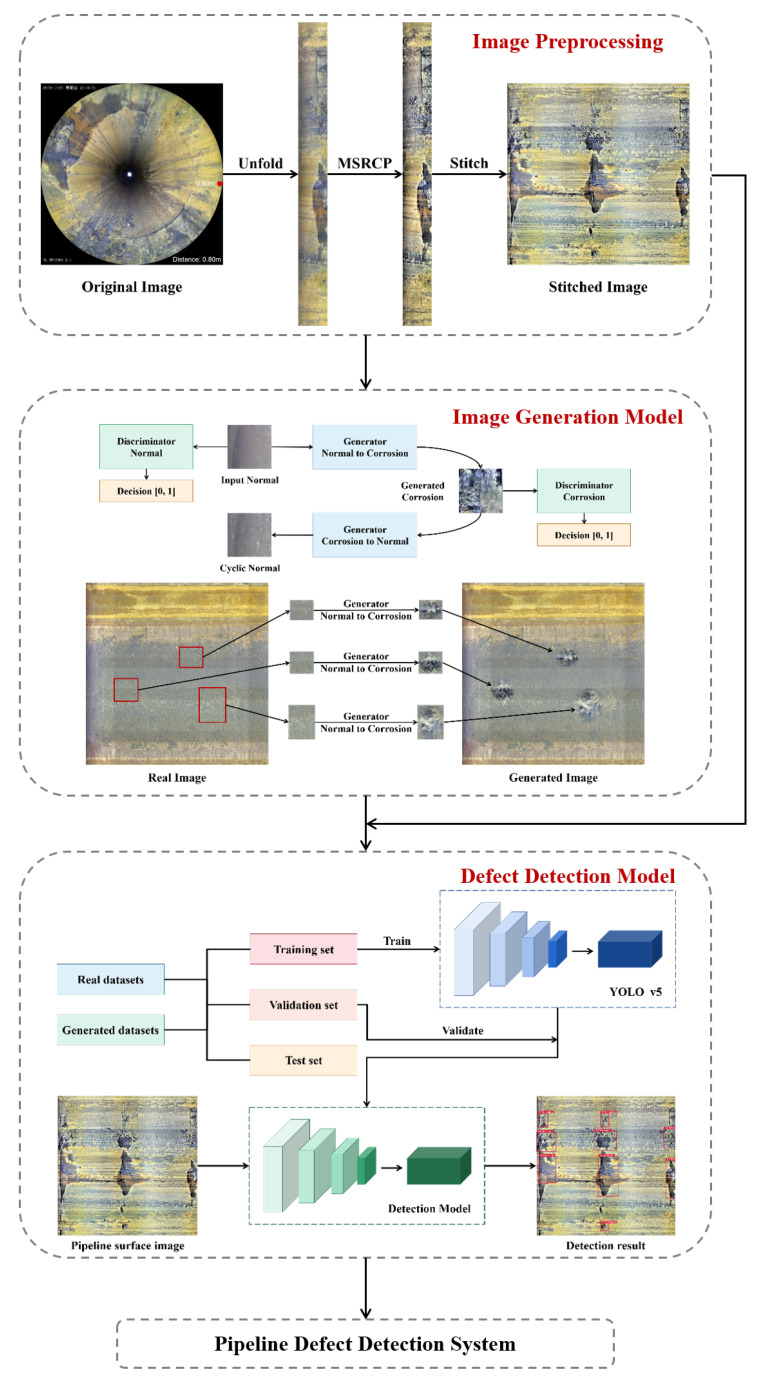
The structure of the defect detection system.

**Figure 2 sensors-22-07907-f002:**
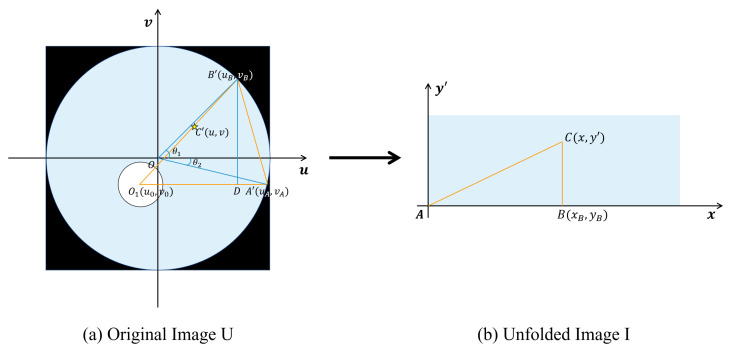
Original image and unfolded image.

**Figure 3 sensors-22-07907-f003:**
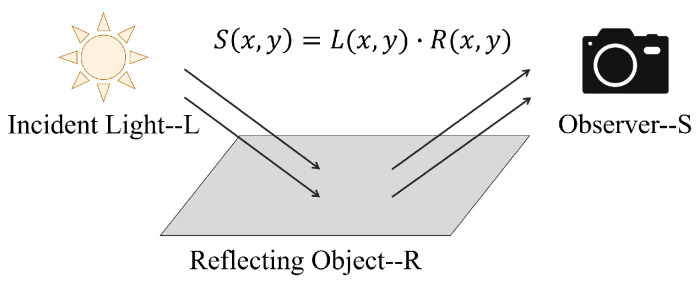
Composition of images in Retinex theory.

**Figure 4 sensors-22-07907-f004:**
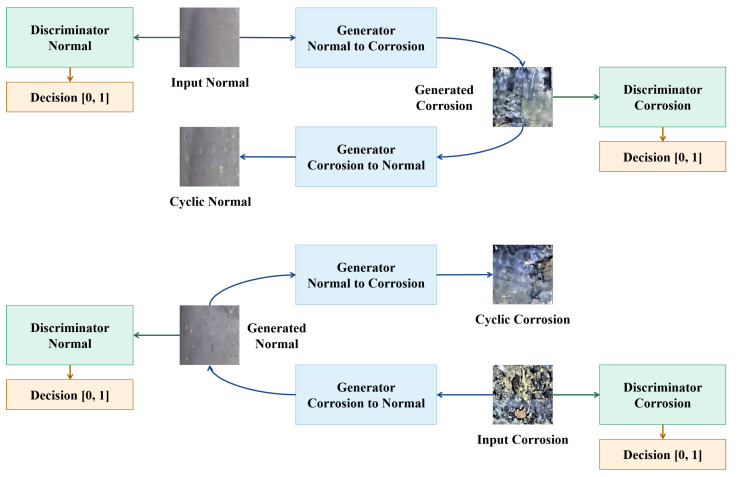
The flowchart of the cycle-GAN network.

**Figure 5 sensors-22-07907-f005:**
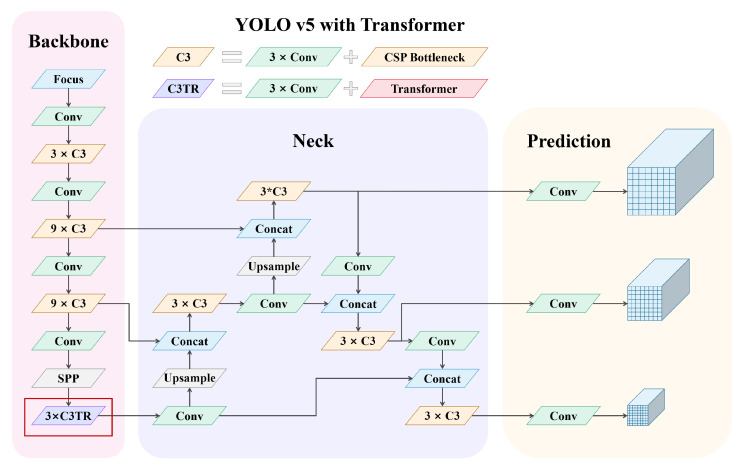
The network structure of YOLO v5.

**Figure 6 sensors-22-07907-f006:**
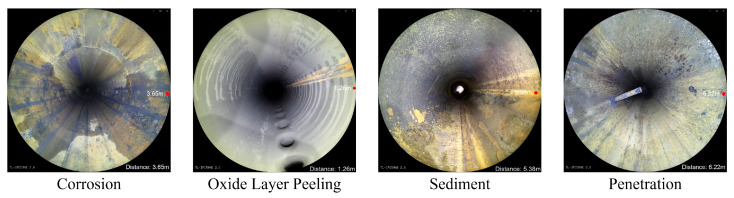
Examples of different defect categories.

**Figure 7 sensors-22-07907-f007:**
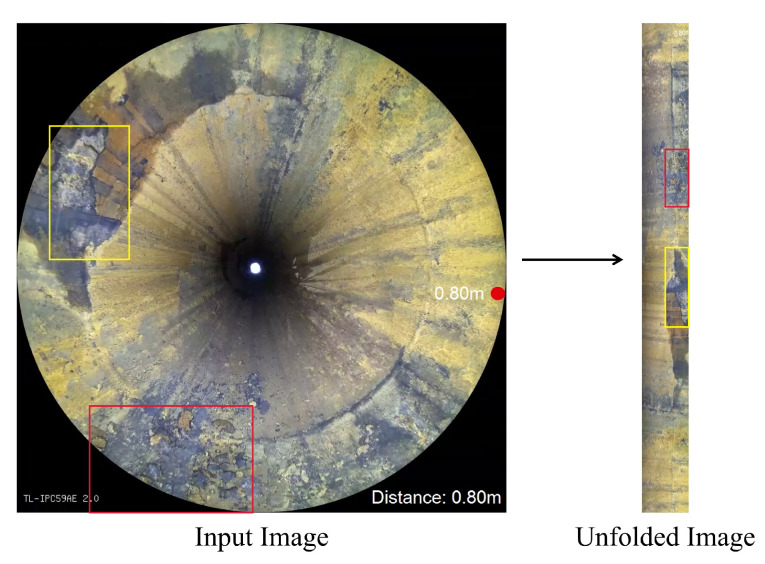
Unfolding results.

**Figure 8 sensors-22-07907-f008:**
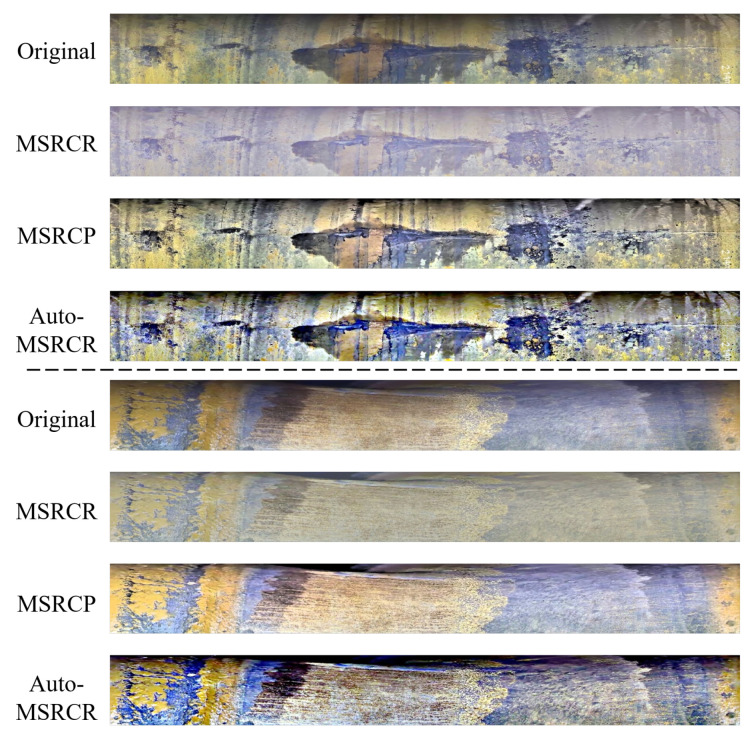
Comparison of the effects of different image enhancement methods.

**Figure 9 sensors-22-07907-f009:**
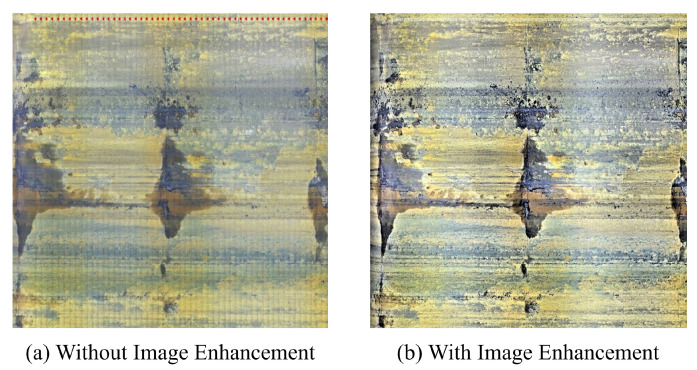
Image stitching results.

**Figure 10 sensors-22-07907-f010:**
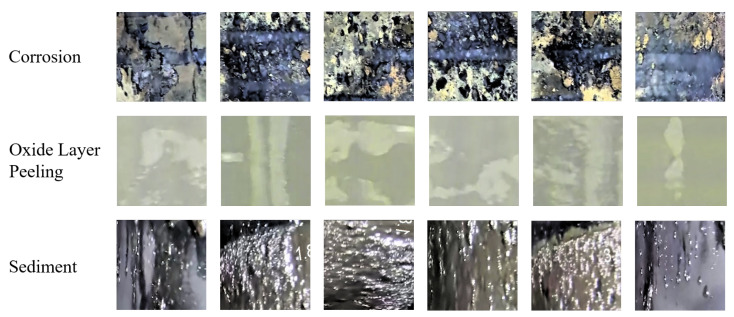
Local images of corrosion, oxide layer peeling, and sediment.

**Figure 11 sensors-22-07907-f011:**
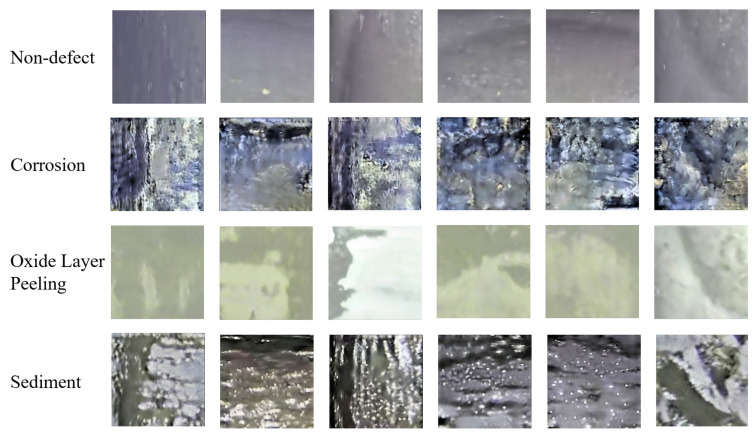
The non-defect images in the original pipeline and the results generated by Cycle-GAN.

**Figure 12 sensors-22-07907-f012:**
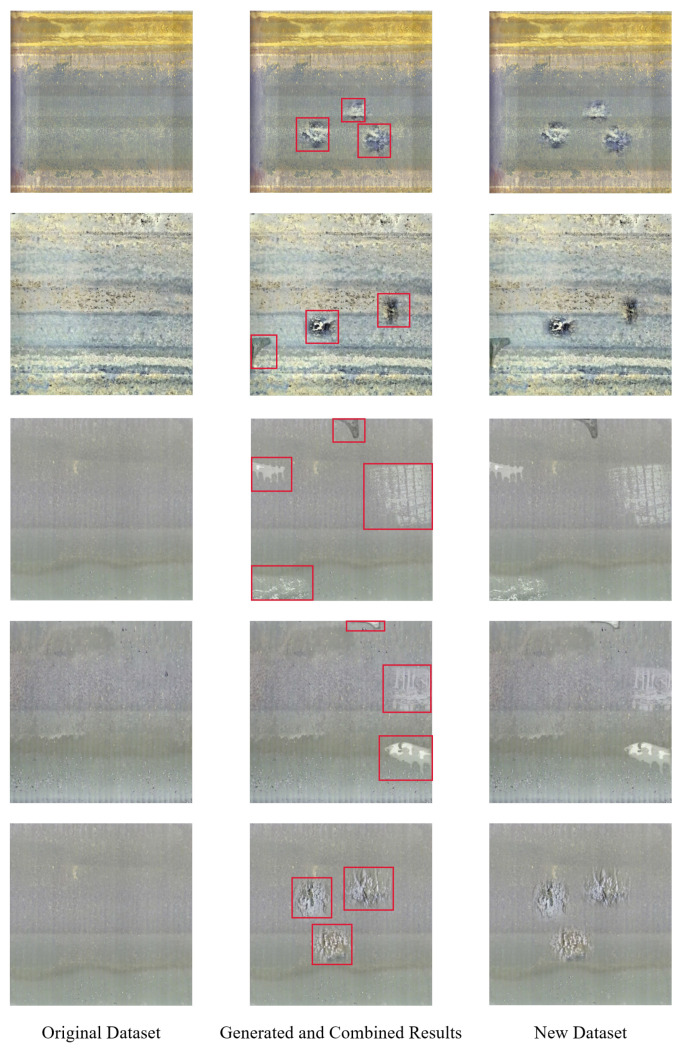
The original images and the fusion results.

**Figure 13 sensors-22-07907-f013:**
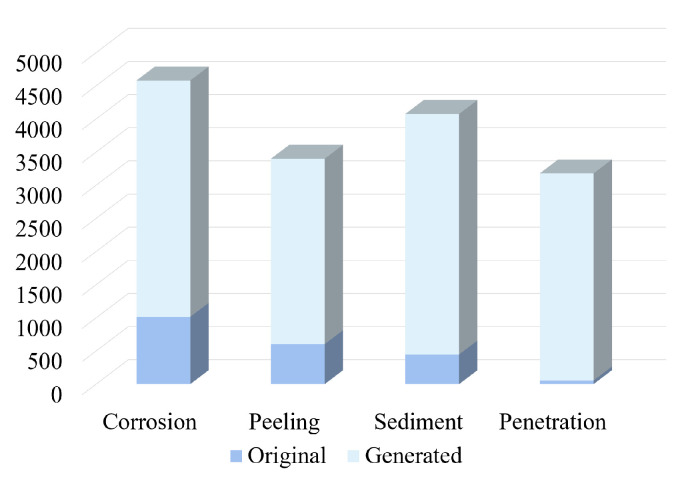
Distribution ratio graph of original and generated samples for each type of defect.

**Figure 14 sensors-22-07907-f014:**
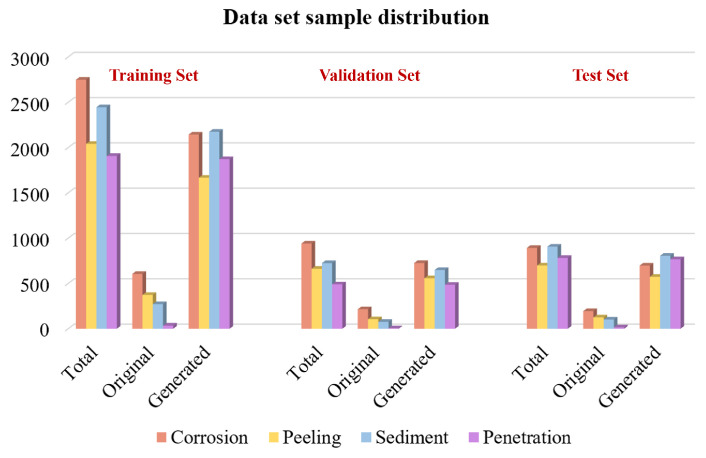
Proportion of sample distribution in the training set, validation set, and test set.

**Figure 15 sensors-22-07907-f015:**
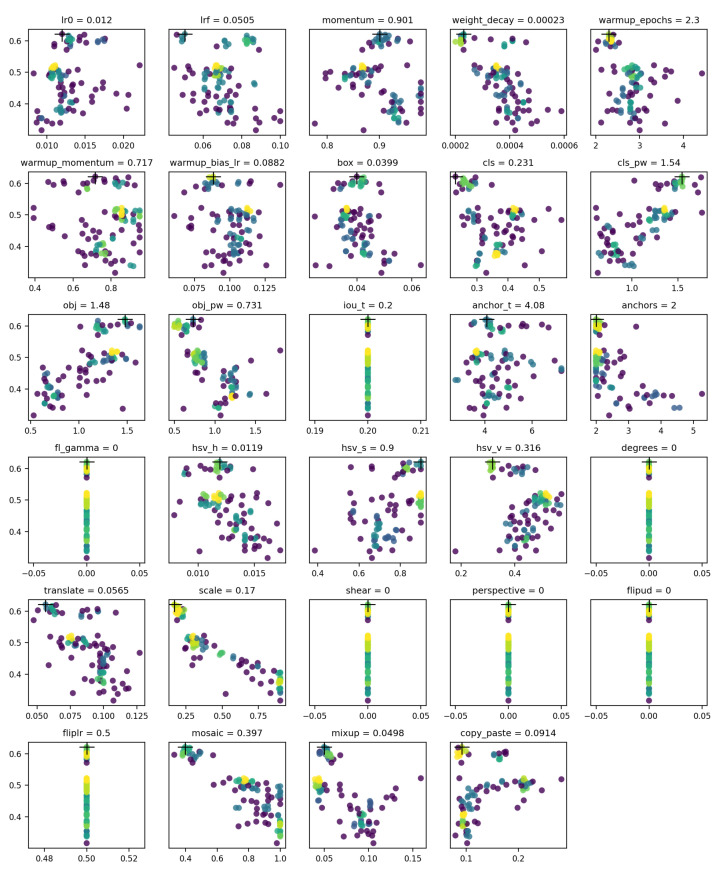
Hyperparameter evolution process and final hyperparameter settings.

**Figure 16 sensors-22-07907-f016:**
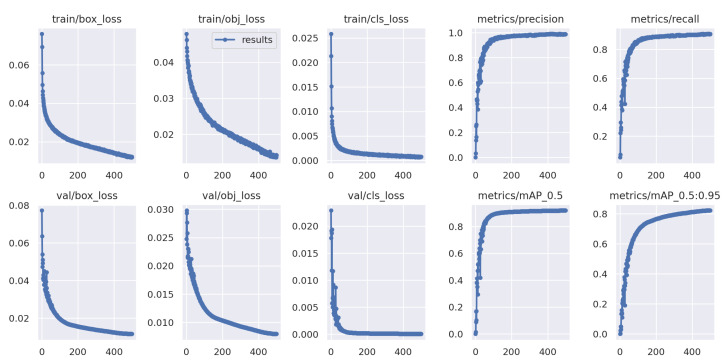
Loss function and training metrics.

**Figure 17 sensors-22-07907-f017:**
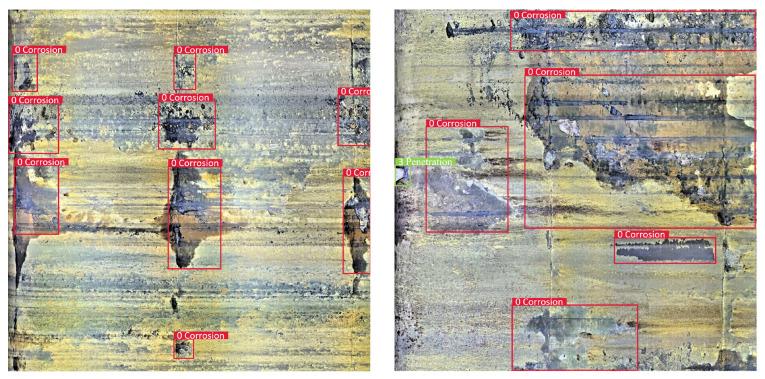
Detection result on the distillation pipeline.

**Figure 18 sensors-22-07907-f018:**
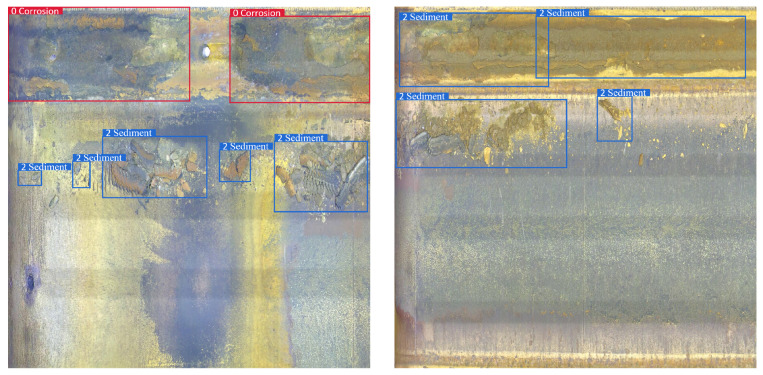
Detection result on the catalytic pipeline.

**Figure 19 sensors-22-07907-f019:**
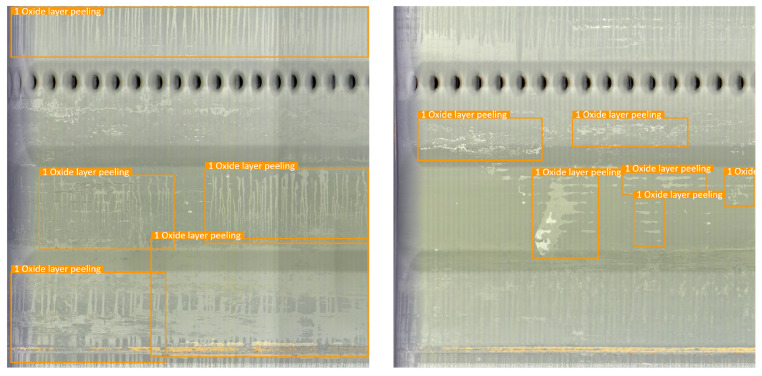
Detection result on the reforming collection pipeline.

**Figure 20 sensors-22-07907-f020:**
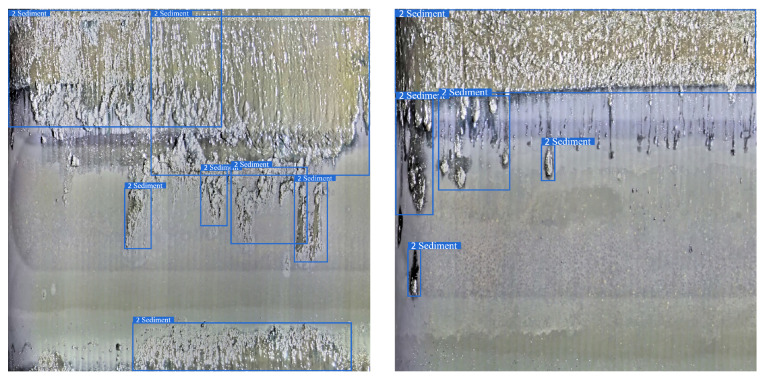
Detection result on the drainage pipeline.

**Table 1 sensors-22-07907-t001:** Results of the pipeline defect detection model based on Cycle-GAN and YOLO v5.

Predicted Number	Ground-Truth Number						
	Corrosion	Peeling	Sediment	Penetration	Background	Total	Precision
Corrosion	813	13	33	17	5	881	92.28%
Peeling	21	627	25	12	7	692	90.61%
Sediment	26	27	817	21	3	894	91.39%
Penetration	3	5	6	725	0	739	98.11%
Background	29	26	25	7			
Total	892	698	906	782			93.10%
Recall	91.14%	89.83%	90.18%	92.71%		90.96%	
F1-score	0.917	0.91	0.912	0.925		0.920	

**Table 2 sensors-22-07907-t002:** Results of the defect detection model on original images.

Predicted Number	Ground-Truth Number						
	Corrosion	Peeling	Sediment	Penetration	Background	Total	Precision
Corrosion	174	3	6	0	5	188	92.55%
Peeling	7	110	2	0	6	125	88.00%
Sediment	9	3	89	1	1	103	86.41%
Penetration	1	0	0	14	0	15	93.33%
Background	4	9	4	0			
Total	195	125	101	15			90.07%
Recall	89.23%	88.00%	88.12%	93.33%		89.67%	
F1-score	0.909	0.902	0.903	0.929		0.899	

**Table 3 sensors-22-07907-t003:** Results of the defect detection model on generated images.

Predicted Number	Ground-Truth Number						
	Corrosion	Peeling	Sediment	Penetration	Background	Total	Precision
Corrosion	639	10	27	17	0	693	92.21%
Peeling	14	517	23	12	1	567	91.18%
Sediment	17	24	728	20	2	791	92.04%
Penetration	2	5	6	711	0	724	98.20%
Background	25	17	21	7			
Total	697	573	805	767			93.41%
Recall	91.68%	90.23%	90.43%	92.70%		91.26%	
F1-score	0.919	0.912	0.913	0.925		0.923	

**Table 4 sensors-22-07907-t004:** Defect detection result based on original images.

Predicted Number	Ground-Truth Number						
	Corrosion	Peeling	Sediment	Penetration	Background	Total	Precision
Corrosion	314	17	25	4	15	375	83.73%
Peeling	5	384	3	1	23	416	92.31%
Sediment	26	19	163	5	9	222	73.42%
Penetration	5	6	7	83	3	104	79.81%
Background	6	23	10	2			
Total	356	449	208	95			82.32%
Recall	88.20%	85.52%	78.37%	87.37%		84.86%	
F1-score	0.859	0.846	0.81	0.855		0.836	

**Table 5 sensors-22-07907-t005:** Defect detection result based on unstitched images.

Predicted Number	Ground-Truth Number						
	Corrosion	Peeling	Sediment	Penetration	Background	Total	Precision
Corrosion	319	13	19	2	11	364	87.64%
Peeling	4	397	4	0	19	424	93.63%
Sediment	22	18	172	5	13	230	74.78%
Penetration	3	4	6	86	1	100	86.00%
Background	8	17	7	2			
Total	356	449	208	95			85.51%
Recall	89.61%	88.42%	82.69%	90.53%		87.81%	
F1-score	0.886	0.88	0.851	0.891		0.866	

**Table 6 sensors-22-07907-t006:** Defect detection results of three preprocessing methods.

	Original	Unfold	Ours (Unfold & Stitch)
Precision	82.32%	85.51%	93.10%
Recall	84.86%	87.81%	90.96%
F1-score	0.836	0.866	0.92

**Table 7 sensors-22-07907-t007:** Defect detection result without data enhancement.

Predicted Number	Ground-Truth Number						
	Corrosion	Peeling	Sediment	Penetration	Background	Total	Precision
Corrosion	15	1	2	0	6	24	62.50%
Peeling	3	7	0	0	4	14	50.00%
Sediment	6	1	6	1	3	17	35.29%
Penetration	1	0	1	1	1	4	25.00%
Background	4	4	2	0			
Total	29	13	11	2			43.20%
Recall	51.72%	53.85%	54.55%	50.00%		52.53%	
F1-score	0.566	0.579	0.583	0.556		0.474	

**Table 8 sensors-22-07907-t008:** Defect detection results of YOLO v5 without attention mechanism.

Predicted Number	Ground-Truth Number						
	Corrosion	Peeling	Sediment	Penetration	Background	Total	Precision
Corrosion	799	19	39	26	7	890	89.78%
Peeling	23	597	47	10	11	688	86.77%
Sediment	31	37	799	29	4	900	88.78%
Penetration	5	6	9	703	2	725	96.97%
Background	34	39	12	14			
Total	892	698	906	782			90.57%
Recall	89.57%	85.53%	88.19%	89.90%		88.30%	
F1-score	0.897	0.876	0.89	0.898		0.894	

**Table 9 sensors-22-07907-t009:** Defect detection results of YOLO v5 with CBAM.

Predicted Number	Ground-Truth Number						
	Corrosion	Peeling	Sediment	Penetration	Background	Total	Precision
Corrosion	805	17	38	21	7	888	90.65%
Peeling	27	609	41	12	11	700	87.00%
Sediment	29	33	803	21	4	890	90.22%
Penetration	4	7	5	711	2	729	97.53%
Background	27	32	19	17			
Total	892	698	906	782			91.35%
Recall	90.25%	87.25%	88.63%	90.92%		89.26%	
F1-score	0.904	0.889	0.896	0.908		0.903	

**Table 10 sensors-22-07907-t010:** Defect detection results of YOLO v5 with SE.

Predicted Number	Ground-Truth Number						
	Corrosion	Peeling	Sediment	Penetration	Background	Total	Precision
Corrosion	809	21	33	27	6	896	90.29%
Peeling	23	603	49	9	14	698	86.39%
Sediment	31	29	811	25	3	899	90.21%
Penetration	4	8	4	709	3	728	97.39%
Background	25	37	9	12			
Total	892	698	906	782			91.07%
Recall	90.70%	86.39%	89.51%	90.66%		89.32%	
F1-score	0.905	0.883	0.899	0.905		0.902	

**Table 11 sensors-22-07907-t011:** Defect detection results of different attention mechanisms.

	None	CBAM	SE	Ours (Transformer)
Precision	90.57%	91.35%	91.07	93.10%
Recall	88.30%	89.26%	89.32%	90.96%
F1-score	0.894	0.903	0.902	0.92

## Data Availability

Not applicable.

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
