# Peer review of "An Automatic Defect Detection System for Petrochemical Pipeline Based on Cycle-GAN and YOLO v5"

_sensors, 2022, doi:10.3390/s22207907_

Round 1

Reviewer 1 Report

This paper presented an automatic detection system for petrochemical pipeline defects based on cyclic GAN and improved YOLO v5. Aiming at the problem of small sample size and unbalanced defect and non-defect categories, a sample enhancement strategy based on cyclic GAN is proposed to generate defect images and expand the data set. In order to improve the detection accuracy, a robust defect detection model based on improved YOLO v5 and Transformer attention mechanism is proposed.

The paper is well organized, and the result is meaningful. The following comments are provided for the author to revise the paper.

(1) The key to the deep learning method for pipeline defect recognition is how to convert the original pipeline video into a more regular picture, that is, how to create a data set. It can be seen from Section 5.1 that different image preprocessing methods have a significant impact on defect identification. Therefore, is there a standard process for pipeline image preprocessing? Please provide additional information.

(2) The paper compares three attention methods: SE (channel attention), CBAM (channel+spatial attention), and Transformer. There is a comparison of relevant methods in the computer field, so is it necessary to compare the three in the paper?

(3) The proportion of training, verification, and test sets mentioned in the article are 8:1:1. The detailed distribution of the original data set and the new samples generated by the Cycle GAN network in this proportion is not described in detail. Whether the test set only contains the original samples or whether the original samples are mixed with the new samples generated is recommended to be supplemented.

(4) This paper uses the Cycle GAN network to generate pseudo samples. How to ensure that the generated pseudo samples are credible and can be used for the subsequent defect detection when the original samples are few? Explanations should be included.

Reviewer 2 Report

Comments:

The topic is very interesting. “An Automatic Defect Detection System for Petrochemical Pipeline Based On Cycle-GAN and YOLO v5”.

However, in the present form, some issues need clarity. For improving the quality of the paper, the authors can address the following comments:

1.      The problem statement is clear and the objective of the study is to propose an automatic defect detection system for petrochemical pipelines. This was based on n Cycle-GAN and improved YOLO v5. But, the methodology of the proposed system is not clear in the abstract. The authors can revise the methodology part in the abstract to make it clear. Also, a briefing on the findings can be stated in the abstract.

2.      The authors praise on Closed Circuit Television (CCTV) system in the introduction section. Therefore, a comparison between the proposed system and the mentioned system needed to be developed in the discussion to view the new contribution or superiority.

3.      Figure 1 is not clear as in figure 5 and why it is repeated in figure 5?

4.      The related work can be one section and an extensive literature review and argument between the previous methods can be stated. Please improve in section 2.

5.      Typos errors and English writing can be improved.

6.      Reference can be looking to high-impact journals and the latest 3 years on the topic. Please improve references.

Round 2

Reviewer 2 Report

All concerns and comments raised are addressed.

Suggestion: The paper content is too much; the authors can improve and remove the unnecessary parts thought by authors.